# Distributionally Robust Graphical Models

**Rizal Fathony, Ashkan Rezaei, Mohammad Ali Bashiri, Xinhua Zhang, Brian D. Ziebart**
Department of Computer Science, University of Illinois at Chicago
Chicago, IL 60607
{rfatho2, arezae4, mbashi4, zhangx, bziebart}@uic.edu

## Abstract

In many structured prediction problems, complex relationships between variables are compactly defined using graphical structures. The most prevalent graphical prediction methods—probabilistic graphical models and large margin methods—have their own distinct strengths but also possess significant drawbacks. Conditional random fields (CRFs) are Fisher consistent, but they do not permit integration of customized loss metrics into their learning process. Large-margin models, such as structured support vector machines (SSVMs), have the flexibility to incorporate customized loss metrics, but lack Fisher consistency guarantees. We present adversarial graphical models (AGM), a distributionally robust approach for constructing a predictor that performs robustly for a class of data distributions defined using a graphical structure. Our approach enjoys both the flexibility of incorporating customized loss metrics into its design as well as the statistical guarantee of Fisher consistency. We present exact learning and prediction algorithms for AGM with time complexity similar to existing graphical models and show the practical benefits of our approach with experiments.

## 1 Introduction

Learning algorithms must consider complex relationships between variables to provide useful predictions in many structured prediction problems. These complex relationships are often represented using graphs to convey the independence assumptions being employed. For example, chain structures are used when modeling sequences like words and sentences [1], tree structures are popular for natural language processing tasks that involve prediction for entities in parse trees [2–4], and lattice structures are often used for modeling images [5]. The most prevalent methods for learning with graphical structure are probabilistic graphical models (e.g., conditional random fields (CRFs) [6]) and large margin models (e.g., structured support vector machines (SSVMs) [7] and maximum margin Markov networks ($M^3$Ns) [8]). Both types of models have unique advantages and disadvantages. CRFs with sufficiently expressive feature representation are consistent estimators of the marginal probabilities of variables in cliques of the graph [9], but are oblivious to the evaluative loss metric during training. On the other hand, SSVMs directly incorporate the evaluative loss metric in the training optimization, but lack consistency guarantees for multiclass settings [10, 11].

To address these limitations, we propose adversarial graphical models (AGM), a distributionally robust framework for leveraging graphical structure among variables that provides both the flexibility to incorporate customized loss metrics during training as well as the statistical guarantee of Fisher consistency for a chosen loss metric. Our approach is based on a robust adversarial formulation [12–14] that seeks a predictor that minimizes a loss metric in the worst-case given the statistical summaries of the empirical distribution. We replace the empirical training data for evaluating our predictor with an adversary that is free to choose an evaluating distribution from the set of distributions that match the statistical summaries of empirical training data via moment matching constraints, as defined by a graphical structure.

Our AGM framework accepts a variety of loss metrics. A notable example that connects our framework to previous models is the logarithmic loss metric. The conditional random field (CRF) model [6] can be viewed as the robust predictor that best minimizes the logarithmic loss metric in the worst case subject to moment matching constraints. In this paper, we focus on a family of loss matrices that additively decomposes over each variable and is defined only based on the label values of the predictor and evaluator. For examples, the additive zero-one (the Hamming loss), ordinal regression (absolute), and cost sensitive metrics fall into this family of loss metrics. We propose efficient exact algorithms for learning and prediction for graphical structures with low treewidth. Finally, we experimentally demonstrate the benefits of our framework compared with the previous models on structured prediction tasks.

## 2 Background and related works

### 2.1 Structured prediction, Fisher consistency, and graphical models

The structured prediction task is to simultaneously predict correlated label variables $\mathbf{y} \in \mathcal{Y}$—often given input variables $\mathbf{x} \in \mathcal{X}$—to minimize a loss metric (e.g., loss $: \mathcal{Y} \times \mathcal{Y} \to \mathbb{R}$) with respect to the true label values $\tilde{\mathbf{y}}$. This is in contrast with classification methods that predict one single variable $y$. Given a distribution over the multivariate labels, $P(\mathbf{y})$, **Fisher consistency** is a desirable characteristic that requires a learning method to produce predictions $\hat{\mathbf{y}}$ that minimize the expected loss of this distribution, $\hat{\mathbf{y}}^* \in \operatorname{argmin}_{\hat{\mathbf{y}}} \mathbb{E}_{\mathbf{Y} \sim \tilde{P}}[\operatorname{loss}(\hat{\mathbf{y}}, \mathbf{Y})]$, under ideal learning conditions (i.e., trained from the true data distribution using a fully expressive feature representation).

To reduce the complexity of the mappings from $\mathcal{X}$ to $\mathcal{Y}$ being learned, independence assumptions and more restrictive representations are employed. In probabilistic graphical models, such as Bayesian networks [15] and random fields [6], these assumptions are represented using a graph over the variables. For graphs with arbitrary structure, inference (i.e., computing posterior probabilities or maximal value assignments) requires exponential time in terms of the number of variables [16]. However, this run-time complexity reduces to be polynomial in terms of the number of predicted variables for graphs with low treewidth (e.g., chains, trees, cycles).

### 2.2 Conditional random fields as robust multivariate log loss minimization

Following ideas from robust Bayes decision theory [12, 13] and distributional robustness [17], the conditional random field [6] can be derived as a robust minimizer of the logarithmic loss subject to moment-matching constraints:

$$\min_{\hat{P}(\cdot|\mathbf{x})} \max_{\check{P}(\cdot|\mathbf{x})} \mathbb{E}_{\substack{\mathbf{X} \sim \tilde{\mathbf{P}}; \\ \check{\mathbf{Y}}|\mathbf{X} \sim \check{P}}} \left[ -\log \hat{P}(\check{\mathbf{Y}}|\mathbf{X}) \right] \text{ such that: } \mathbb{E}_{\substack{\mathbf{X} \sim \tilde{\mathbf{P}}; \\ \check{\mathbf{Y}}|\mathbf{X} \sim \check{P}}} \left[ \Phi(\mathbf{X}, \check{\mathbf{Y}}) \right] = \mathbb{E}_{\mathbf{X},\mathbf{Y} \sim \tilde{P}} \left[ \Phi(\mathbf{X}, \mathbf{Y}) \right], \quad (1)$$

where $\Phi : \mathcal{X} \times \mathcal{Y} \to \mathbb{R}^k$ are feature functions that typically decompose additively over subsets of variables. Under this perspective, the predictor $\hat{P}$ seeks the conditional distribution that minimizes log loss against an adversary $\check{P}$ seeking to choose an evaluation distribution that approximates training data statistics, while otherwise maximizing log loss. As a result, the predictor is robust not only to the training sample $\tilde{P}$, but all distributions with matching moment statistics [13].

The saddle point for Eq. (1) is obtained by the parametric conditional distribution $\hat{P}_\theta(\mathbf{y}|\mathbf{x}) = \check{P}_\theta(\mathbf{y}|\mathbf{x}) = e^{\theta \cdot \Phi(\mathbf{x},\mathbf{y})} / \sum_{\mathbf{y}' \in \mathcal{Y}} e^{\theta \cdot \Phi(\mathbf{x},\mathbf{y}')}$ with parameters $\theta$ chosen by maximizing the data likelihood: $\operatorname{argmax}_\theta \mathbb{E}_{\mathbf{X},\mathbf{Y} \sim \tilde{P}} \left[ \log \hat{P}_\theta(\mathbf{Y}|\mathbf{X}) \right]$. The decomposition of the feature function into additive clique features, $\Phi_i(\mathbf{x}, \mathbf{y}) = \sum_{c \in \mathcal{C}_i} \phi_{c,i}(\mathbf{x}_c, \mathbf{y}_c)$, can be represented graphically by connecting the variables within cliques with undirected edges. Dynamic programming algorithms (e.g., junction tree) allow the exact likelihood to be computed in run time that is exponential in terms of the treewidth of the resulting graph [18].

Predictions for a particular loss metric are then made using the Bayes optimal prediction for the estimated distribution: $\mathbf{y}^* = \operatorname{argmin}_{\mathbf{y}} \mathbb{E}_{\hat{\mathbf{Y}}|\mathbf{x} \sim \hat{P}_\theta}[\operatorname{loss}(\mathbf{y}, \hat{\mathbf{Y}})]$. This two-stage prediction approach can create inefficiencies when learning from limited amounts of data since optimization may focus on accurately estimating probabilities in portions of the input space that have no impact on the decision boundaries of the Bayes optimal prediction. Rather than separating the prediction task from the

learning process, we incorporate the evaluation loss metric of interest into the robust minimization formulation of Eq. (1) in this work.

## 2.3 Structured support vector machines

Our approach is most similar to structured support vector machines (SSVMs) [19] and related maximum margin methods [8], which also directly incorporate the evaluation loss metric into the training process. This is accomplished by minimizing a hinge loss convex surrogate:

$$\text{hinge}_\theta(\tilde{\mathbf{y}}) = \max_{\mathbf{y}} \left\{ \text{loss}(\mathbf{y}, \tilde{\mathbf{y}}) + \theta \cdot \big( \Phi(\mathbf{x}, \tilde{\mathbf{y}}) - \Phi(\mathbf{x}, \mathbf{y}) \big) \right\}, \tag{2}$$

where $\theta$ represents the model parameters, $\tilde{\mathbf{y}}$ is the ground truth label, and $\Phi(\mathbf{x}, \mathbf{y})$ is a feature function that decomposes additively over subsets of variables.

Using a clique-based graphical representation of the potential function, and assuming the loss metric also additively decomposes into the same clique-based representation, SSVMs have a computational complexity similar to probabilistic graphical models. Specifically, finding the value assignment $\mathbf{y}$ that maximizes this loss-augmented potential can be accomplished using dynamic programming in run time that is exponential in the graph treewidth [18].

A key weakness of support vector machines in general is their lack of Fisher consistency; there are distributions for multiclass prediction tasks for which the SVM will not learn a Bayes optimal predictor, even when the models are given access to the true distribution and sufficiently expressive features, due to the disconnection between the Crammer-Singer hinge loss surrogate [20] and the evaluation loss metric (i.e., the 0-1 loss in this case) [11]. In practice, if the empirical data behaves similarly to those distributions (e.g., $P(\mathbf{y}|\mathbf{x})$ have no majority $\mathbf{y}$ for a specific input $\mathbf{x}$), the inconsistent model may perform poorly. This inconsistency extends to the structured prediction setting except in limited special cases [21]. We overcome these theoretical deficiencies in our approach by using an adversarial formulation that more closely aligns the training objective with the evaluation loss metric, while maintaining convexity.

## 2.4 Other related works

**Distributionally robust learning**. There has been a recent surge of interest in the machine learning community for developing distributonally robust learning algorithms. The proposed learning algorithms differ in the uncertainty sets used to provide robustness. Previous robust learning algorithms have been proposed under the F-divergence measures (which includes the popular KL-divergence and $\chi$-divergence) [22–24], the Wasserstein metric uncertainty set [25–27], and the moment matching uncertainty set [17, 28]. Our robust adversarial learning approach differs from the previous approaches by focusing on the robustness in terms of the conditional distribution $P(\mathbf{y}|\mathbf{x})$ instead of the joint distribution $P(\mathbf{x}, \mathbf{y})$. Our approach seeks a predictor that is robust to the worst-case conditional label probability under the moment matching constraints. We do not impose any robustness to the training examples $\mathbf{x}$.

**Robust adversarial formulation**. There have been some investigations on applying robust adversarial formulations for prediction to specific types of structured prediction problems (e.g., sequence tagging [29], and graph cuts [30]). Our work differs from them in two key aspects: we provide a general framework for graphical structures and any additive loss metrics, as opposed to the specific structures and loss metrics (additive zero-one loss) previously considered; and we also propose a learning algorithm with polynomial run-time guarantees, in contrast with previously employed algorithms that use double/single oracle constraint generation techniques [31] lacking polynomial run-time guarantees.

**Consistent methods**. A notable research interest in consistent methods for structured prediction tasks has also been observed. Ciliberto et al. [32] proposed a consistent regularization approach that maps the original structured prediction problem into a kernel Hilbert space and employs a multivariate regression on the Hilbert space. Osokin et al. [33] proposed a consistent quadratic surrogate for any structured prediction loss metric and provide a polynomial sample complexity analysis for the additive zero-one loss metric surrogate. Our work differs from these line of works in the focus on the structure. We focus on the graphical structures that model interaction between labels, whereas the previous works focus on the structure of the loss metric itself.

# 3 Approach

We propose adversarial graphical models (AGMs) to better align structured prediction with evaluation loss metrics in settings where the structured interaction between labels are represented in a graph.

## 3.1 Formulations

We construct a predictor that best minimizes a loss metric for the worst-case evaluation distribution that (approximately) matches the statistical summaries of empirical training data. Our predictor is allowed to make a probabilistic prediction over all possible label assignments (denoted as $\hat{P}(\hat{\mathbf{y}}|\mathbf{x})$). However, instead of evaluating the prediction with empirical data (as commonly performed by empirical risk minimization formulations [34]), the predictor is pitted against an adversary that also makes a probabilistic prediction (denoted as $\check{P}(\check{\mathbf{y}}|\mathbf{x})$). The adversary is constrained to select its conditional distributions to match the statistical summaries of the empirical training distribution (denoted as $\tilde{P}$) via moment matching constraints on the features functions $\Phi$.

**Definition 1.** *The adversarial prediction method for structured prediction problems with graphical interaction between labels is:*

$$\min_{\hat{P}(\hat{\mathbf{y}}|\mathbf{x})} \max_{\check{P}(\check{\mathbf{y}}|\mathbf{x})} \mathbb{E}_{\substack{\mathbf{X}\sim\tilde{P};\\ \hat{\mathbf{Y}}|\mathbf{X}\sim\hat{P};\\ \check{\mathbf{Y}}|\mathbf{X}\sim\check{P}}} \left[ loss(\hat{\mathbf{Y}}, \check{\mathbf{Y}}) \right] \text{ such that: } \mathbb{E}_{\substack{\mathbf{X}\sim\tilde{P};\\ \check{\mathbf{Y}}|\mathbf{X}\sim\check{P}}} \left[ \Phi(\mathbf{X}, \check{\mathbf{Y}}) \right] = \tilde{\Phi}, \tag{3}$$

*where the vector of feature moments, $\tilde{\Phi} = \mathbb{E}_{\mathbf{X},\mathbf{Y}\sim\tilde{P}}[\Phi(\mathbf{X}, \mathbf{Y})]$, is measured from sample training data. The feature function $\Phi(\mathbf{X}, \mathbf{Y})$ contains features that are additively decomposed over cliques in the graph, e.g. $\Phi(\mathbf{x}, \mathbf{y}) = \sum_c \phi(\mathbf{x}, \mathbf{y}_c)$.*

This follows recent research in developing adversarial prediction for cost-sensitive classification [14] and multivariate performance metrics [35], and, more generally, distributionally robust decision making under moment-matching constraints [17]. In this paper, we focus on pairwise graphical structures where the interactions between labels are defined over the edges (and nodes) of the graph. We also restrict the loss metric to a family of metrics that additively decompose over each $y_i$ variable, i.e., $loss(\hat{\mathbf{y}}, \check{\mathbf{y}}) = \sum_{i=1}^n loss(\hat{y}_i, \check{y}_i)$. Directly solving the optimization in Eq. (3) is impractical for reasonably-sized problems since $P(\mathbf{y}|\mathbf{x})$ grows exponentially with the number of predicted variables. Instead, we utilize the method of Lagrange multipliers and the marginal formulation of the distributions of predictor and adversary to formulate a simpler dual optimization problem as stated in Theorem 1.

**Theorem 1.** *For the adversarial structured prediction with pairwise graphical structure and an additive loss metric, solving the optimization in Definition 1 is equivalent to solving the following expectation of maximin problems over the node and edge marginal distributions parameterized by Lagrange multipiers $\theta$:*

$$\min_{\theta_e,\theta_v} \mathbb{E}_{\mathbf{X},\mathbf{Y}\sim\tilde{P}} \max_{\check{P}(\check{\mathbf{y}}|\mathbf{x})} \min_{\hat{P}(\hat{\mathbf{y}}|\mathbf{x})} \left[ \sum_i^n \sum_{\hat{y}_i,\check{y}_i} \hat{P}(\hat{y}_i|\mathbf{x})\check{P}(\check{y}_i|\mathbf{x})loss(\hat{y}_i, \check{y}_i) \right. \tag{4}$$
$$+ \sum_{(i,j)\in E} \sum_{\check{y}_i,\check{y}_j} \check{P}(\check{y}_i,\check{y}_j|\mathbf{x}) \left[ \theta_e \cdot \phi(\mathbf{x}, \check{y}_i, \check{y}_j) \right] - \sum_{(i,j)\in E} \theta_e \cdot \phi(\mathbf{x}, y_i, y_j)$$
$$\left. + \sum_i^n \sum_{\check{y}_i} \check{P}(\check{y}_i|\mathbf{x}) \left[ \theta_v \cdot \phi(\mathbf{x}, \check{y}_i) \right] - \sum_i^n \theta_v \cdot \phi(\mathbf{x}, y_i) \right],$$

*where $\phi(\mathbf{x}, y_i)$ is the node feature function for node $i$, $\phi(\mathbf{x}, y_i, y_j)$ is the edge feature function for the edge connecting node $i$ and $j$, $E$ is the set of edges in the graphical structure, and $\theta_v$ and $\theta_e$ are the Lagrange dual variables for the moment matching constraints corresponding to the node and edge features, respectively. The optimization objective depends on the predictor's probability prediction $\hat{P}(\hat{\mathbf{y}}|\mathbf{x})$ only through its node marginal probabilities $\hat{P}(\hat{y}_i|\mathbf{x})$. Similarly, the objective depends on the adversary's probabilistic prediction $\check{P}(\check{\mathbf{y}}|\mathbf{x})$ only through its node and edge marginal probabilities, i.e., $\check{P}(\check{y}_i|\mathbf{x})$, and $\check{P}(\check{y}_i, \check{y}_j|\mathbf{x})$.*

*Proof sketch.* From Eq. (3) we use the method of Langrange multipliers to introduce dual variables $\theta_v$ and $\theta_e$ that represent the moment-matching constraints over node and edge features, respectively. Using the strong duality theorem for convex-concave saddle point problems [36, 37], we swap the

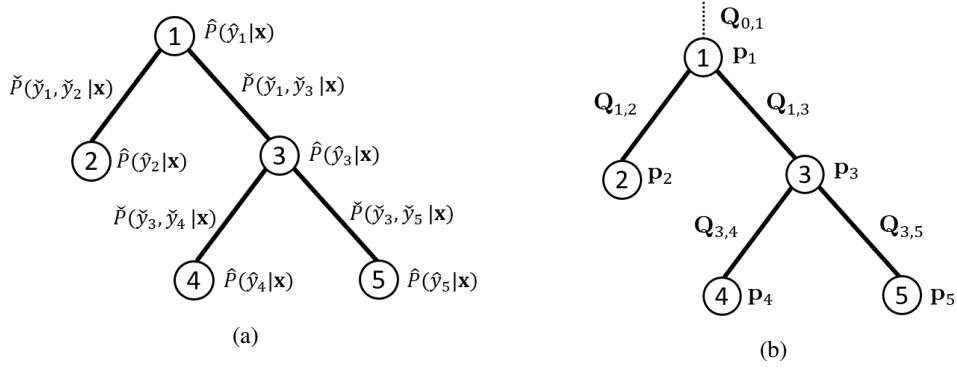

Figure 1: An example tree structure with five nodes and four edges with the corresponding marginal probabilities for predictor and adversary (a); and the matrix and vector notations of the probabilities (b). Note that we introduce a dummy variable $\mathbf{Q}_{0,1}$ to match the constraints in Eq. (5).

optimization order of $\theta$, $\hat{P}(\hat{\mathbf{y}}|\mathbf{x})$, and $\hat{P}(\check{\mathbf{y}}|\mathbf{x})$ as in Eq. (4). Then, using the additive property of the loss metric and feature functions, the optimization over $\hat{P}(\hat{\mathbf{y}}|\mathbf{x})$ and $\hat{P}(\check{\mathbf{y}}|\mathbf{x})$ can be transformed into an optimization over their respective node and edge marginal distributions.[1] $\qquad\square$

Note that the optimization in Eq. (4) over the node and edge marginal distributions resembles the optimization of CRFs [38]. In terms of computational complexity, this means that for a general graphical structure, the optimization above may be intractable. We focus on families of graphical structures in which the optimization is known to be tractable. In the next subsection, we begin with the case of tree-structured graphical models and then proceed with the case of graphical models with low treewidth. In both cases, we formulate the corresponding efficient learning algorithms.

## 3.2 Optimization

We first introduce our vector and matrix notations for AGM optimization. Without loss of generality, we assume the number of class labels $k$ to be the same for all predicted variables $y_i, \forall i \in \{1, \ldots, n\}$. Let $\mathbf{p}_i$ be a vector with length $k$, where its $a$-th element contains $\hat{P}(\hat{y}_i = a|\mathbf{x})$, and let $\mathbf{Q}_{i,j}$ be a $k$-by-$k$ matrix with its $(a,b)$-th cells store $\check{P}(\check{y}_i = a, \check{y}_j = b|\mathbf{x})$. We also use a vector and matrix notation to represent the ground truth label by letting $\mathbf{z}_i$ be a one-hot vector where its $a$-th element $\mathbf{z}_i^{(a)} = 1$ if $y_i = a$ or otherwise 0, and letting $\mathbf{Z}_{i,j}$ be a one-hot matrix where its $(a,b)$-th cell $\mathbf{Z}_{i,j}^{(a,b)} = 1$ if $y_i = a \wedge y_j = b$ or otherwise 0. For each node feature $\phi_l(\mathbf{x}, y_i)$, we denote $\mathbf{w}_{i,l}$ as a length $k$ vector where its $a$-th element contains the value of $\phi_l(\mathbf{x}, y_i = a)$. Similarly, for each edge feature $\phi_l(\mathbf{x}, y_i, y_j)$, we denote $\mathbf{W}_{i,j,l}$ as a $k$-by-$k$ matrix where its $(a,b)$-th cell contains the value of $\phi_l(\mathbf{x}, y_i = a, y_j = b)$. For a pairwise graphical model with tree structure, we rewrite Eq. (4) using our vector and matrix notation with local marginal consistency constraints as follows:

$$\min_{\theta_e, \theta_v} \mathbb{E}_{\mathbf{X},\mathbf{Y} \sim \tilde{P}} \max_{\mathbf{Q} \in \Delta} \min_{\mathbf{p} \in \Delta} \sum_{i}^{n} \left[ \mathbf{p}_i \mathbf{L}_i (\mathbf{Q}_{pt(i);i}^{\mathrm{T}} \mathbf{1}) + \left\langle \mathbf{Q}_{pt(i);i} - \mathbf{Z}_{pt(i);i}, \textstyle\sum_l \theta_e^{(l)} \mathbf{W}_{pt(i);i;l} \right\rangle \right. \tag{5}$$

$$\left. + (\mathbf{Q}_{pt(i);i}^{\mathrm{T}} \mathbf{1} - \mathbf{z}_i)^{\mathrm{T}} (\textstyle\sum_l \theta_v^{(l)} \mathbf{w}_{i;l}) \right]$$

$$\text{subject to: } \mathbf{Q}_{pt(pt(i));pt(i)}^{\mathrm{T}} \mathbf{1} = \mathbf{Q}_{pt(i);i} \mathbf{1}, \ \forall i \in \{1, \ldots, n\},$$

where $pt(i)$ indicates the parent of node $i$ in the tree structure, $\mathbf{L}_i$ stores a loss matrix corresponding to the portion of the loss metric for node $i$, i.e., $\mathbf{L}_i^{(a,b)} = \text{loss}(\hat{y}_i = a, \check{y}_i = b)$, and $\langle \cdot, \cdot \rangle$ denotes the Frobenius inner product between two matrices, i.e., $\langle A, B \rangle = \sum_{i,j} A_{i,j} B_{i,j}$. Note that we also employ probability simplex constraints ($\Delta$) to each $\mathbf{Q}_{pt(i);i}$ and $\mathbf{p}_i$. Figure 1 shows an example tree structure with its marginal probabilities and the matrix notation of the probabilities.

### 3.2.1 Learning algorithm

We first focus on solving the inner minimax optimization of Eq. (5). To simplify our notation, we denote the edge potentials $\mathbf{B}_{pt(i);i} = \sum_l \theta_e^{(l)} \mathbf{W}_{pt(i);i;l}$ and the node potentials $\mathbf{b}_i = \sum_l \theta_v^{(l)} \mathbf{w}_{i;l}$. We then rewrite the inner optimization of Eq. (5) as:

$$\max_{\mathbf{Q}\in\Delta} \min_{\mathbf{p}\in\Delta} \sum_i^n \left[ \mathbf{p}_i \mathbf{L}_i (\mathbf{Q}_{pt(i);i}^{\mathrm{T}} \mathbf{1}) + \left\langle \mathbf{Q}_{pt(i);i}, \mathbf{B}_{pt(i);i} \right\rangle + (\mathbf{Q}_{pt(i);i}^{\mathrm{T}} \mathbf{1})^{\mathrm{T}} \mathbf{b}_i \right] \tag{6}$$

$$\text{subject to: } \mathbf{Q}_{pt(pt(i));pt(i)}^{\mathrm{T}} \mathbf{1} = \mathbf{Q}_{pt(i);i} \mathbf{1}, \ \forall i \in \{1, \dots, n\}.$$

To solve the optimization above, we use dual decomposition technique [39, 40] that decompose the dual version of the optimization problem into several sub-problem that can be solved independently. By introducing the Lagrange variable $\mathbf{u}$ for the local marginal consistency constraint, we formulate an equivalent dual unconstrained optimization problem as shown in Theorem 2.

**Theorem 2.** *The constrained optimization in Eq.* (6) *is equivalent to an unconstrained Lagrange dual problem with an inner optimization that can be solved independently for each node as follows:*

$$\min_{\mathbf{u}} \sum_i^n \left[ \max_{\mathbf{Q}_i\in\Delta} \left\langle \mathbf{Q}_{pt(i);i}, \mathbf{B}_{pt(i);i} + \mathbf{1}\mathbf{b}_i^{\mathrm{T}} - \mathbf{u}_i \mathbf{1}^{\mathrm{T}} + \sum_{k\in ch(i)} \mathbf{1}\mathbf{u}_k^{\mathrm{T}} \right\rangle + \min_{\mathbf{p}_i\in\Delta} \mathbf{p}_i \mathbf{L}_i (\mathbf{Q}_{pt(i);i}^{\mathrm{T}} \mathbf{1}) \right], \tag{7}$$

*where $\mathbf{u}_i$ is the Lagrange dual variable associated with the marginal constraint of $\mathbf{Q}_{pt(pt(i));pt(i)}^{\mathrm{T}} \mathbf{1} = \mathbf{Q}_{pt(i);i} \mathbf{1}$, and $ch(i)$ represent the children of node $i$.*

*Proof sketch.* From Eq. (6) we use the method of Langrange multipliers to introduce dual variables $\mathbf{u}$. The resulting dual optimization admits dual decomposability, where we can rearrange the variables into independent optimizations for each node as in Eq. (7). □

We denote matrix $\mathbf{A}_{pt(i);i} \triangleq \mathbf{B}_{pt(i);i} + \mathbf{1}\mathbf{b}_i^{\mathrm{T}} - \mathbf{u}_i \mathbf{1}^{\mathrm{T}} + \sum_{k\in ch(i)} \mathbf{1}\mathbf{u}_k^{\mathrm{T}}$ to simplify the inner optimization in Eq. (7). Let us define $\mathbf{r}_i \triangleq \mathbf{Q}_{pt(i);i}^{\mathrm{T}} \mathbf{1}$ and $\mathbf{a}_i$ be the column wise maximum of matrix $\mathbf{A}_{pt(i);i}$, i.e., $\mathbf{a}_i^{(l)} = \max_l \mathbf{A}_{l;i}$. Given the value of $\mathbf{u}$, each of the inner optimizations in Eq. (7) can be equivalently solved in terms of our newly defined variable changes $\mathbf{r}_i$ and $\mathbf{a}_i$ as follows:

$$\max_{\mathbf{r}_i\in\Delta} \left[ \mathbf{a}_i^{\mathrm{T}} \mathbf{r}_i + \min_{\mathbf{p}_i\in\Delta} \mathbf{p}_i \mathbf{L}_i \mathbf{r}_i \right]. \tag{8}$$

Note that this resembles the optimization in a standard adversarial multiclass classification problem [14] with $\mathbf{L}_i$ as the loss matrix and $\mathbf{a}_i$ as the class-based potential vector. For an arbitrary loss matrix, Eq. (8) can be solved as a linear program. However, this is somewhat slow and more efficient algorithms have been studied in the case of zero-one and ordinal regression metrics [41, 42]. Given the solution of this inner optimization, we use a sub-gradient based optimization to find the optimal Lagrange dual variables $\mathbf{u}^*$.

To recover our original variables for the adversary's marginal distribution $\mathbf{Q}_{pt(i);i}^*$ given the optimal dual variables $\mathbf{u}^*$, we use the following steps. First, we use $\mathbf{u}^*$ and Eq. (8) to compute the value of the node marginal probability $\mathbf{r}_i^*$. With the additional information that we know the value of $\mathbf{r}_i^*$ (i.e., the adversary's node probability), Eq. (6) can be solved independently for each $\mathbf{Q}_{pt(i);i}$ to obtain the optimal $\mathbf{Q}_{pt(i);i}^*$ as follows:

$$\mathbf{Q}_{pt(i);i}^* = \underset{\mathbf{Q}_{pt(i);i}\in\Delta}{\operatorname{argmax}} \left\langle \mathbf{Q}_{pt(i);i}, \mathbf{B}_{pt(i);i} \right\rangle \text{ subject to: } \mathbf{Q}_{pt(i);i}^{\mathrm{T}} \mathbf{1} = \mathbf{r}_i^*, \mathbf{Q}_{pt(i);i} \mathbf{1} = \mathbf{r}_{pt(i)}^*. \tag{9}$$

Note that the optimization above resembles an optimal transport problem over two discrete distributions [43] with cost matrix $-\mathbf{B}_{pt(i);i}$. This optimal transport problem can be solved using a linear program solver or a more sophisticated solver (e.g., using Sinkhorn distances [44]).

For our overall learning algorithm, we use the optimal adversary's marginal distributions $\mathbf{Q}_{pt(i);i}^*$ to compute the sub-differential of the AGM formulation (Eq. (5)) with respect to $\theta_v$ and $\theta_e$. The sub-differential for $\theta_v^{(l)}$ includes the expected node feature difference $\mathbb{E}_{\mathbf{X},\mathbf{Y}\sim\tilde{P}} \sum_i^n (\mathbf{Q}_{pt(i);i}^{*\mathrm{T}} \mathbf{1} - \mathbf{z}_i)^{\mathrm{T}} \mathbf{w}_{i;l}$, whereas the sub-differential for $\theta_e^{(l)}$ includes the expected edge feature difference $\mathbb{E}_{\mathbf{X},\mathbf{Y}\sim\tilde{P}} \sum_i^n \left\langle \mathbf{Q}_{pt(i);i}^* - \mathbf{Z}_{pt(i);i}, \mathbf{W}_{pt(i);i;l} \right\rangle$. Using this sub-differential information, we employ a stochastic sub-gradient based algorithm to obtain the optimal $\theta_v^*$ and $\theta_e^*$.

### 3.2.2 Prediction algorithms

We propose two different prediction schemes: probabilistic and non-probabilistic prediction.

**Probabilistic prediction**. Our probabilistic prediction is based on the predictor's label probability distribution in the adversarial prediction formulation. Given fixed values of $\theta_v$ and $\theta_e$, we solve a minimax optimization similar to Eq. (5) by flipping the order of the predictor and adversary distribution as follows:

$$\min_{\mathbf{p}\in\Delta} \max_{\mathbf{Q}\in\Delta} \sum_i^n \left[ \mathbf{p}_i \mathbf{L}_i(\mathbf{Q}_{pt(i);i}^{\mathrm{T}}\mathbf{1}) + \left\langle \mathbf{Q}_{pt(i);i}, \sum_l \theta_e^{(l)}\mathbf{W}_{pt(i);i;l} \right\rangle + (\mathbf{Q}_{pt(i);i}^{\mathrm{T}}\mathbf{1})^{\mathrm{T}}(\sum_l \theta_v^{(l)}\mathbf{w}_{i;l}) \right]$$

$$\text{subject to: } \mathbf{Q}_{pt(pt(i));pt(i)}^{\mathrm{T}}\mathbf{1} = \mathbf{Q}_{pt(i);i}\mathbf{1}, \ \forall i \in \{1,\ldots,n\}. \tag{10}$$

To solve the inner maximization of $\mathbf{Q}$ we use a similar technique as in MAP inference for CRFs. We then use a projected gradient optimization technique to solve the outer minimization over $\mathbf{p}$ and a technique for projecting to the probability simplex [45].

**Non-probabilistic prediction**. Our non-probabilistic prediction scheme is similar to SSVM's prediction algorithm. In this scheme, we find $\hat{\mathbf{y}}$ that maximizes the potential value, i.e., $\hat{\mathbf{y}} = \operatorname{argmax}_{\mathbf{y}} f(\mathbf{x},\mathbf{y})$, where $f(\mathbf{x},\mathbf{y}) = \theta^{\mathrm{T}}\Phi(\mathbf{x},\mathbf{y})$. This prediction scheme is faster than the probabilistic scheme since we only need a single run of a Viterbi-like algorithm for tree structures.

### 3.2.3 Runtime analysis

Each stochastic update in our algorithm involves finding the optimal $\mathbf{u}$ and recovering the optimal $\mathbf{Q}$ to be used in a sub-gradient update. Each iteration of a sub-gradient based optimization to solve $\mathbf{u}$ costs $\mathcal{O}(n \cdot c(\mathbf{L}))$ time where $n$ is the number of nodes and $c(\mathbf{L})$ is the cost for solving the optimization in Eq. (8) for the loss matrix $\mathbf{L}$. Recovering all of the adversary's marginal distributions $\mathbf{Q}_{pt(i);i}$ using a fast Sinkhorn distance solver has the empirical complexity of $\mathcal{O}(nk^2)$ where $k$ is the number of classes [44]. The total running time of our method depends on the loss metric we use. For example, if the loss metric is the additive zero-one loss, the total complexity of one stochastic gradient update is $\mathcal{O}(nlk\log k + nk^2)$ time, where $l$ is the number of iterations needed to obtain the optimal $\mathbf{u}$ and $\mathcal{O}(k\log k)$ time is the cost for solving Eq. (8) for the zero-one loss [41]. In practice, we find the average value of $l$ to be relatively small. This runtime complexity is competitive with the CRF, which requires $\mathcal{O}(nk^2)$ time to perform message-passing over a tree to compute the marginal distribution of each parameter update, and also with structured SVM where each iteration requires computing the most violated constraint, which also costs $\mathcal{O}(nk^2)$ time for running a Viterbi-like algorithm over a tree structure.

### 3.2.4 Learning algorithm for graphical structure with low treewidth

Our algorithm for tree-based graphs can be easily extended to the case of graphical structures with low treewidth. Similar to the case of the junction tree algorithm for probabilistic graphical models, we first construct a junction tree representation for the graphical structure. We then solve a similar optimization as in Eq. (5) on the junction tree. In this case, the time complexity of one stochastic gradient update of the algorithm is $\mathcal{O}(nlwk^{(w+1)}\log k + nk^{2(w+1)})$ time for the optimization with an additive zero-one loss metric, where $n$ is the number of cliques in the junction tree, $k$ is the number of classes, $l$ is the number of iterations in the inner optimization, and $w$ is the treewidth of the graph. This time complexity is competitive with the time complexities of CRF and SSVM which are also exponential in the treewidth of the graph.

### 3.3 Fisher consistency analysis

A key theoretical advantage of our approach over the structured SVM is that it provides Fisher consistency. This guarantees that under the true distribution $P(\mathbf{x},\mathbf{y})$, the learning algorithm yields a Bayes optimal prediction with respect to the loss metric [10, 11]. In this setting, the learning algorithm is allowed to optimize over all measurable functions, or similarly, it has a feature representation of unlimited richness. We establish the Fisher consistency of our AGM approach in Theorem 3.

**Theorem 3.** *The AGM approach is Fisher consistent for all additive loss metrics.*

*Proof.* As established in Theorem 1, pairwise marginal probabilities are sufficient statistics of the adversary's distribution. An unlimited access to arbitrary rich feature representation constrains the adversary's distribution in Eq. (3) to match the marginal probabilities of the true distribution, making the optimization in Eq. (3) equivalent to $\min_{\hat{\mathbf{y}}} \mathbb{E}_{\mathbf{X},\mathbf{Y} \sim P}\big[\text{loss}(\hat{\mathbf{y}}, \mathbf{Y})\big]$, which is the Bayes optimal prediction for the loss metric. □

## 4 Experimental evaluation

To evaluate our approach, we apply AGM to two different tasks: predicting emotion intensity from a sequence of images, and labeling entities in parse trees with semantic roles. We show the benefit of our method compared with a conditional random field (CRF) and a structured SVM (SSVM).

### 4.1 Facial emotion intensity prediction

We evaluate our approach in the facial emotion intensity prediction task [46]. Given a sequence of facial images, the task is to predict the emotion intensity for each individual image. The emotion intensity labels are categorized into three ordinal categories: `neutral <` `increasing < apex`, reflecting the degree of intensity. The dataset contains 167 sequences collected from 100 subjects consisting of six types of basic emotions (anger, disgust, fear, happiness, sadness, and surprise). In terms of the features used for prediction, we follow an existing feature extraction procedure [46] that uses Haar-like features and the PCA algorithm to reduce the feature dimensionality.

Table 1: The average loss metrics for the emotion intensity prediction. Bold numbers indicate the best or not significantly worse than the best results (Wilcoxon signed-rank test with $\alpha = 0.05$).

| Loss metrics | AGM | CRF | SSVM |
|---|---|---|---|
| zero-one, unweighted | 0.34 | **0.32** | 0.37 |
| absolute, unweighted | **0.33** | 0.34 | 0.40 |
| squared, unweighted | **0.38** | **0.38** | 0.40 |
| zero-one, weighted | **0.28** | 0.32 | 0.29 |
| absolute, weighted | **0.29** | 0.36 | **0.29** |
| squared, weighted | 0.36 | 0.40 | **0.33** |
| average | 0.33 | 0.35 | 0.35 |
| # bold | 4 | 2 | 2 |

In our experimental setup, we combine the data from all six different emotions and focus on predicting the ordinal category of emotion intensity. From the whole 167 sequences, we construct 20 different random splits of the training and the testing datasets with 120 sequences of training samples and 47 sequences of testing samples. We use the training set in the first split to perform cross validation to obtain the best regularization parameters and then use the best parameter in the evaluation phase for all 20 different splits of the dataset.

In the evaluation, we use six different loss metrics. The first three metrics are the average of zero-one, absolute and squared loss metrics for each node in the graph (where we assign label values: `neutral = 1`, `increasing = 2`, and `apex = 3`). The other three metrics are the weighted version of the zero-one, absolute and squared loss metrics. These weighted variants of the loss metrics reflect the focus on the prediction task by emphasizing the prediction on particular nodes in the graph. In this experiment, we set the weight to be the position in the sequence so that we focus more on the latest nodes in the sequences.

We compare our method with CRF and SSVM models. Both the AGM and the SSVM can incorporate the task's customized loss metrics in the learning process. The prediction for AGM and SSVM is done by taking an arg-max of potential values, i.e., $\text{argmax}_{\mathbf{y}} f(\mathbf{x}, \mathbf{y}) = \theta \cdot \Phi(\mathbf{x}, \mathbf{y})$. For CRF, the training step aims to model the conditional probability $\hat{P}_\theta(\mathbf{y}|\mathbf{x})$. The CRF's predictions are computed using the Bayes optimal prediction with respect to the loss metric and CRF's conditional probability, i.e., $\text{argmin}_{\mathbf{y}} \mathbb{E}_{\hat{\mathbf{Y}}|\mathbf{x} \sim \hat{P}_\theta}[\text{loss}(\mathbf{y}, \hat{\mathbf{Y}})]$.

We report the loss metrics averaged over the dataset splits as shown in Table 1. We highlight the result that is either the best result or not significantly worse than the best result (using Wilcoxon signed-rank test with $\alpha = 0.05$). The result shows that our method significantly outperforms CRF in three cases (absolute, weighted zero-one, and weighted absolute losses), and statistically ties with CRF in one case (squared loss), while only being outperformed by CRF in one case (zero-one loss). AGM also outperforms SSVM in three cases (absolute, squared, and weighted zero-one losses), and statistically ties with SSVM in one case (weighted absolute loss), while only being outperformed by SSVM in one case (weighted squared loss). In the overall result, AGM maintains advantages

compared to CRFs and SSVMs in both the overall average loss and the number of "indistinguishably best" performances on all cases. These results may reflect the theoretical benefit that AGM has over CRF and SSVM mentioned in Section 3 when learning from noisy labels.

## 4.2 Semantic role labeling

We evaluate the performance of our algorithm on the semantic role labeling task for the CoNLL 2005 dataset [47]. Given a sentence and its syntactic parse tree as the input, the task is to recognize the semantic role of each constituent in the sentence as propositions expressed by some target verbs in the sentence. There are a total of 36 semantic roles grouped by their types of: numbered arguments, adjuncts, references of numbered and adjunct arguments, continuation of each class type and the verb. We prune the syntactic trees according to Xue and Palmer [48], i.e., we only include siblings of the nodes which are on the path from the verb (V) to the root and also the immediate children in case that the node is a propositional phrase (PP). Following the setup used by Cohn and Blunsom [49], we extract the same syntactic and contextual features and label non-argument constituents and children nodes of arguments as "outside" (O).

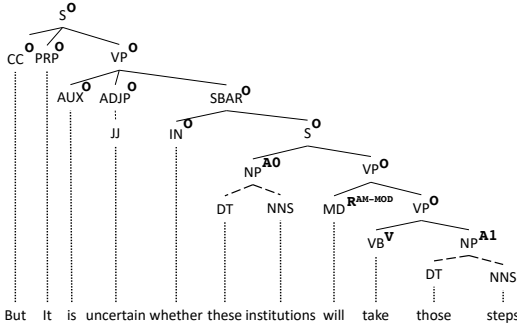

Figure 2: Example of a syntax tree with semantic role labels as bold superscripts. The dotted and dashed lines show the pruned edges from the tree. The original label AM-MOD is among class R in our experimental setup.

Additionally, in our experiment we simplify the prediction task by reducing the number of labels. Specifically, we choose the three most common labels in the WSJ test dataset, i.e., A0,A1,A2 and their references R-A0,R-A1,R-A2, and we combine the rest of the classes as one separate class R. Thus, together with outside O and verb V, we have a total of nine classes in our experiment.

In the evaluation, we use a cost-sensitive loss matrix that reflects the importance of each label. We use the same cost-sensitive loss matrix to evaluate the prediction of all nodes in the graph. The cost-sensitive loss matrix is constructed by picking a random order of the class label and assigning an ordinal loss based on the order of

Table 2: The average loss metrics for the semantic role labeling task.

| Loss metrics | AGM | CRF | SSVM |
|---|---|---|---|
| cost-sensitive loss | 0.14 | 0.19 | 0.14 |

the labels. We compare the average cost-sensitive loss metric of our method with the CRF and the SSVM as shown in Table 2. As we can see from the table, our result is competitive with SSVM, while maintaining an advantage over the CRF. This experiment shows that incorporating customized losses into the training process of learning algorithms is important for some structured prediction tasks. Both the AGM and the SSVM are designed to align their learning algorithms with the customized loss metric, whereas CRF can only utilize the loss metric information in its prediction step.

## 5 Conclusion

In this paper, we introduced adversarial graphical models, a robust approach to structured prediction that possesses the main benefits of existing methods: (1) it guarantees the same Fisher consistency possessed by CRFs [6]; (2) it aligns the target loss metric with the learning objective, as in maximum margin methods [19, 8]; and (3) its computational run time complexity is primarily shaped by the graph treewidth, which is similar to both graphical modeling approaches. Our experimental results demonstrate the benefits of this approach on structured prediction tasks with low treewidth.

For more complex graphical structures with high treewidth, our proposed algorithm may not be efficient. Similar to the case of CRFs and SSVMs, approximation algorithms may be needed to solve the optimization in AGM formulations for these structures. In future work, we plan to investigate the optimization techniques and applicable approximation algorithms for general graphical structures.

**Acknowledgement.** This work was supported, in part, by the National Science Foundation under Grant No. 1652530, and by the Future of Life Institute (futureoflife.org) FLI-RFP-AI1 program.

## Footnotes

[1]More detailed proofs for Theorem 1 and 2 are available in the supplementary material.

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
