[Supplementary Material]

# A Supplementary Materials

## A.1 Proof of Theorem 1

*Proof of Theorem 1.*

$$\min_{\hat{P}(\hat{\mathbf{y}}|\mathbf{x})} \max_{\check{P}(\check{\mathbf{y}}|\mathbf{x})} \mathbb{E}_{\mathbf{X}\sim\tilde{P};\hat{\mathbf{Y}}|\mathbf{X}\sim\hat{P};\check{\mathbf{Y}}|\mathbf{X}\sim\check{P}} \left[ \mathrm{loss}(\hat{\mathbf{Y}}, \check{\mathbf{Y}}) \right] \tag{11}$$

$$\text{subject to: } \mathbb{E}_{\mathbf{X}\sim\tilde{P};\check{\mathbf{Y}}|\mathbf{X}\sim\check{P}} \left[ \Phi(\mathbf{X}, \check{\mathbf{Y}}) \right] = \mathbb{E}_{\mathbf{X},\mathbf{Y}\sim\tilde{P}} \left[ \Phi(\mathbf{X}, \mathbf{Y}) \right]$$

$$\overset{(a)}{=} \max_{\check{P}(\check{\mathbf{y}}|\mathbf{x})} \min_{\hat{P}(\hat{\mathbf{y}}|\mathbf{x})} \mathbb{E}_{\mathbf{X}\sim\tilde{P};\hat{\mathbf{Y}}|\mathbf{X}\sim\hat{P};\check{\mathbf{Y}}|\mathbf{X}\sim\check{P}} \left[ \mathrm{loss}(\hat{\mathbf{Y}}, \check{\mathbf{Y}}) \right] \tag{12}$$

$$\text{subject to: } \mathbb{E}_{\mathbf{X}\sim\tilde{P};\check{\mathbf{Y}}|\mathbf{X}\sim\check{P}} \left[ \Phi(\mathbf{X}, \check{\mathbf{Y}}) \right] = \mathbb{E}_{\mathbf{X},\mathbf{Y}\sim\tilde{P}} \left[ \Phi(\mathbf{X}, \mathbf{Y}) \right]$$

$$\overset{(b)}{=} \max_{\check{P}(\check{\mathbf{y}}|\mathbf{x})} \min_{\theta} \min_{\hat{P}(\hat{\mathbf{y}}|\mathbf{x})} \mathbb{E}_{\mathbf{X},\mathbf{Y}\sim\tilde{P};\hat{\mathbf{Y}}|\mathbf{X}\sim\hat{P};\check{\mathbf{Y}}|\mathbf{X}\sim\check{P}} \left[ \mathrm{loss}(\hat{\mathbf{Y}}, \check{\mathbf{Y}}) + \theta^{\mathrm{T}} \left( \Phi(\mathbf{X}, \check{\mathbf{Y}}) - \Phi(\mathbf{X}, \mathbf{Y}) \right) \right] \tag{13}$$

$$\overset{(c)}{=} \min_{\theta} \max_{\check{P}(\check{\mathbf{y}}|\mathbf{x})} \min_{\hat{P}(\hat{\mathbf{y}}|\mathbf{x})} \mathbb{E}_{\mathbf{X},\mathbf{Y}\sim\tilde{P};\hat{\mathbf{Y}}|\mathbf{X}\sim\hat{P};\check{\mathbf{Y}}|\mathbf{X}\sim\check{P}} \left[ \mathrm{loss}(\hat{\mathbf{Y}}, \check{\mathbf{Y}}) + \theta^{\mathrm{T}} \left( \Phi(\mathbf{X}, \check{\mathbf{Y}}) - \Phi(\mathbf{X}, \mathbf{Y}) \right) \right] \tag{14}$$

$$\overset{(d)}{=} \min_{\theta} \mathbb{E}_{\mathbf{X},\mathbf{Y}\sim\tilde{P}} \max_{\check{P}(\check{\mathbf{y}}|\mathbf{x})} \min_{\hat{P}(\hat{\mathbf{y}}|\mathbf{x})} \mathbb{E}_{\hat{\mathbf{Y}}|\mathbf{X}\sim\hat{P};\check{\mathbf{Y}}|\mathbf{X}\sim\check{P}} \left[ \mathrm{loss}(\hat{\mathbf{Y}}, \check{\mathbf{Y}}) + \theta^{\mathrm{T}} \left( \Phi(\mathbf{X}, \check{\mathbf{Y}}) - \Phi(\mathbf{X}, \mathbf{Y}) \right) \right] \tag{15}$$

$$\overset{(e)}{=} \min_{\theta_e,\theta_v} \mathbb{E}_{\mathbf{X},\mathbf{Y}\sim\tilde{P}} \max_{\check{P}(\check{\mathbf{y}}|\mathbf{x})} \min_{\hat{P}(\hat{\mathbf{y}}|\mathbf{x})} \mathbb{E}_{\hat{\mathbf{Y}}|\mathbf{X}\sim\hat{P};\check{\mathbf{Y}}|\mathbf{X}\sim\check{P}} \Big[ \sum_i^n \mathrm{loss}(\hat{Y}_i, \check{Y}_i) \tag{16}$$

$$+ \theta_e \cdot \sum_{(i,j)\in E} \left[ \phi(\mathbf{X}, \check{Y}_i, \check{Y}_j) - \phi(\mathbf{X}, Y_i, Y_j) \right] + \theta_v \cdot \sum_i^n \left[ \phi(\mathbf{X}, \check{Y}_i) - \phi(\mathbf{X}, Y_i) \right] \Big]$$

$$\overset{(f)}{=} \min_{\theta_e,\theta_v} \mathbb{E}_{\mathbf{X},\mathbf{Y}\sim\tilde{P}} \max_{\check{P}(\check{\mathbf{y}}|\mathbf{x})} \min_{\hat{P}(\hat{\mathbf{y}}|\mathbf{x})} \sum_{\hat{\mathbf{y}},\check{\mathbf{y}}} \hat{P}(\hat{\mathbf{y}}|\mathbf{x})\check{P}(\check{\mathbf{y}}|\mathbf{x}) \Big[ \sum_i^n \mathrm{loss}(\hat{y}_i, \check{y}_i) \tag{17}$$

$$+ \theta_e \cdot \sum_{(i,j)\in E} \left[ \phi(\mathbf{x}, \check{y}_i, \check{y}_j) - \phi(\mathbf{x}, y_i, y_j) \right] + \theta_v \cdot \sum_i^n \left[ \phi(\mathbf{x}, \check{y}_i) - \phi(\mathbf{x}, y_i) \right] \Big]$$

$$\overset{(g)}{=} \min_{\theta_e,\theta_v} \mathbb{E}_{\mathbf{X},\mathbf{Y}\sim\tilde{P}} \max_{\check{P}(\check{\mathbf{y}}|\mathbf{x})} \min_{\hat{P}(\hat{\mathbf{y}}|\mathbf{x})} \Big[ \sum_i^n \sum_{\hat{y}_i,\check{y}_i} \hat{P}(\hat{y}_i|\mathbf{x})\check{P}(\check{y}_i|\mathbf{x})\mathrm{loss}(\hat{y}_i, \check{y}_i) \tag{18}$$

$$+ \sum_{(i,j)\in E} \sum_{\check{y}_i,\check{y}_j} \check{P}(\check{y}_i, \check{y}_j|\mathbf{x}) \left[ \theta_e \cdot \phi(\mathbf{x}, \check{y}_i, \check{y}_j) \right] - \sum_{(i,j)\in E} \theta_e \cdot \phi(\mathbf{x}, y_i, y_j)$$

$$+ \sum_i^n \sum_{\check{y}_i} \check{P}(\check{y}_i|\mathbf{x}) \left[ \theta_v \cdot \phi(\mathbf{x}, \check{y}_i) \right] - \sum_i^n \theta_v \cdot \phi(\mathbf{x}, y_i) \Big].$$

The transformation steps above are described as follows:

(a) We flip the min and max order using minimax duality [36]. The domains of $\hat{P}(\hat{\mathbf{y}}|\mathbf{x})$ and $\check{P}(\check{\mathbf{y}}|\mathbf{x})$ are both compact convex sets and the objective function is bilinear, therefore, strong duality holds.

(b) We introduce the Lagrange dual variable $\theta$ to directly incorporate the equality constraints into the objective function.

(c) The domain of $\check{P}(\check{\mathbf{y}}|\mathbf{x})$ is a compact convex subset of $\mathbb{R}^n$, while the domain of $\theta$ is $\mathbb{R}^m$. The objective is concave on $\check{P}(\check{\mathbf{y}}|\mathbf{x})$ for all $\theta$ (a non-negative linear combination of minimums of affine functions is concave), while it is convex on $\theta$ for all $\check{P}(\check{\mathbf{y}}|\mathbf{x})$. Based on Sion's minimax theorem [37], strong duality holds, and thus we can flip the optimization order of $\check{P}(\check{\mathbf{y}}|\mathbf{x})$ and $\theta$.

(d) Since the expression is additive in terms of $\check{P}(\check{\mathbf{y}}|\mathbf{x})$ and $\hat{P}(\hat{\mathbf{y}}|\mathbf{x})$, we can push the expectation over the empirical distribution $\mathbf{X}, Y \sim \tilde{P}$ outside and independently optimize each $\check{P}(\check{\mathbf{y}}|\mathbf{x})$ and $\hat{P}(\hat{\mathbf{y}}|\mathbf{x})$.

(e) We apply our description of loss metrics which is additively decomposable into the loss for each node, and the features that can be decomposed into node and edge features. We also

separate the notation for the Lagrange dual variable into the variable for the constraints on node features ($\theta_v$) and and the variable for the edge features ($\theta_e$).

(f) We rewrite the expectation over $\hat{P}(\hat{\mathbf{y}}|\mathbf{x})$ and $\check{P}(\check{\mathbf{y}}|\mathbf{x})$ in terms of the probability-weighted average.

(g) Based on the property of the loss metrics and feature functions, the sum over the exponentially many possibilities of $\hat{\mathbf{y}}$ and $\check{\mathbf{y}}$ can be simplified into the sum over individual nodes and edges values, resulting in the optimization over the node and edge marginal distributions.

$\square$

## A.2 Proof of Theorem 2

*Proof of Theorem 2.*

$$\max_{\mathbf{Q}\in\Delta}\min_{\mathbf{p}\in\Delta}\sum_i^n \left[ \mathbf{p}_i \mathbf{L}_i(\mathbf{Q}_{pt(i);i}^{\mathrm{T}}\mathbf{1}) + \left\langle \mathbf{Q}_{pt(i);i}, \mathbf{B}_{pt(i);i} \right\rangle + (\mathbf{Q}_{pt(i);i}^{\mathrm{T}}\mathbf{1})^{\mathrm{T}}\mathbf{b}_i \right] \tag{19}$$

$$\text{subject to: } \mathbf{Q}_{pt(pt(i));pt(i)}^{\mathrm{T}}\mathbf{1} = \mathbf{Q}_{pt(i);i}\mathbf{1}, \ \forall i \in \{1,\ldots,n\}$$

$$\overset{(a)}{=} \max_{\mathbf{Q}\in\Delta}\min_{\mathbf{u}}\min_{\mathbf{p}\in\Delta}\sum_i^n \left[ \mathbf{p}_i \mathbf{L}_i(\mathbf{Q}_{pt(i);i}^{\mathrm{T}}\mathbf{1}) + \left\langle \mathbf{Q}_{pt(i);i}, \mathbf{B}_{pt(i);i} \right\rangle + (\mathbf{Q}_{pt(i);i}^{\mathrm{T}}\mathbf{1})^{\mathrm{T}}\mathbf{b}_i \right] \tag{20}$$

$$+ \sum_i^n \mathbf{u}_i^{\mathrm{T}} \left( \mathbf{Q}_{pt(pt(i));pt(i)}^{\mathrm{T}}\mathbf{1} - \mathbf{Q}_{pt(i);i}\mathbf{1} \right)$$

$$\overset{(b)}{=} \min_{\mathbf{u}}\max_{\mathbf{Q}\in\Delta}\min_{\mathbf{p}\in\Delta}\sum_i^n \left[ \mathbf{p}_i \mathbf{L}_i(\mathbf{Q}_{pt(i);i}^{\mathrm{T}}\mathbf{1}) + \left\langle \mathbf{Q}_{pt(i);i}, \mathbf{B}_{pt(i);i} \right\rangle + (\mathbf{Q}_{pt(i);i}^{\mathrm{T}}\mathbf{1})^{\mathrm{T}}\mathbf{b}_i \right] \tag{21}$$

$$+ \sum_i^n \mathbf{u}_i^{\mathrm{T}} \left( \mathbf{Q}_{pt(pt(i));pt(i)}^{\mathrm{T}}\mathbf{1} - \mathbf{Q}_{pt(i);i}\mathbf{1} \right)$$

$$\overset{(c)}{=} \min_{\mathbf{u}}\max_{\mathbf{Q}\in\Delta}\min_{\mathbf{p}\in\Delta}\sum_i^n \left[ \mathbf{p}_i \mathbf{L}_i(\mathbf{Q}_{pt(i);i}^{\mathrm{T}}\mathbf{1}) + \left\langle \mathbf{Q}_{pt(i);i}, \mathbf{B}_{pt(i);i} \right\rangle + \left\langle \mathbf{Q}_{pt(i);i}, \mathbf{1}\mathbf{b}_i^{\mathrm{T}} \right\rangle \right] \tag{22}$$

$$+ \sum_i^n \left[ \left\langle \mathbf{Q}_{pt(pt(i));pt(i)}, \mathbf{1}\mathbf{u}_i^{\mathrm{T}} \right\rangle - \left\langle \mathbf{Q}_{pt(i);i}, \mathbf{u}_i\mathbf{1}^{\mathrm{T}} \right\rangle \right]$$

$$\overset{(d)}{=} \min_{\mathbf{u}}\max_{\mathbf{Q}\in\Delta}\min_{\mathbf{p}\in\Delta}\sum_i^n \left[ \mathbf{p}_i \mathbf{L}_i(\mathbf{Q}_{pt(i);i}^{\mathrm{T}}\mathbf{1}) + \left\langle \mathbf{Q}_{pt(i);i}, \mathbf{B}_{pt(i);i} + \mathbf{1}\mathbf{b}_i^{\mathrm{T}} - \mathbf{u}_i\mathbf{1}^{\mathrm{T}} + \sum_{k\in ch(i)} \mathbf{1}\mathbf{u}_k^{\mathrm{T}} \right\rangle \right].$$

$$\tag{23}$$

The transformation steps above are described as follows:

(a) We introduce the Lagrange dual variable $\mathbf{u}$, where $\mathbf{u}_i$ is the dual variable associated with the marginal constraint of $\mathbf{Q}_{pt(pt(i));pt(i)}^{\mathrm{T}}\mathbf{1} = \mathbf{Q}_{pt(i);i}\mathbf{1}$.

(b) Similar to the analysis in Theorem 1, strong duality holds due to Sion's minimax theorem. Therefore we can flip the optimization order of $\mathbf{Q}$ and $\mathbf{u}$.

(c) We rewrite the vector multiplication over $\mathbf{Q}_{pt(i);i}\mathbf{1}$ or $\mathbf{Q}_{pt(i);i}^{\mathrm{T}}\mathbf{1}$ with the corresponding Frobenius inner product notations.

(d) We regroup the terms in the optimization above by considering the parent-child relations in the tree for each node. Note that $ch(i)$ represents the children of node $i$.

$\square$