[Reviews · NeurIPS 2018]

Reviewer 1



A nice paper about a new class of ‘adversarial’ conditional random field able to embed customized loss functions. Learning and inference algorithms for tree topologies to be extended to the low-treewidth case are proposed. Experimental results show a clear improvement with respect to CRFs and competitive performance with structured SVMs. I think the results are sounds and the paper might be accepted. Yet, I missed some efforts about the motivating Fisher consistency as a desirable features even from a practical perspective. Perhaps evaluating the gap between AGMs and SSVMs for different training set sizes can do that.

Reviewer 2



Distributionally Robust Graphical Models The authors suggest dealing with the structure prediction task using adversarial graphical model (AGM), a generative model trained using an adversary distribution, instead of the empirical one. Instead of focusing on the loss metric, the authors focus on the graphical model allowing more flexibility wrt the loss metric. The AGM algorithm has similar complexity to conditional random fields but is less limited in its loss metric and is Fisher consistent for additive loss metrics. Following a complex mathematical transformation, the authors provide optimization of node and edge distribution of the graphical model but since this optimization is intractable, they restrict their method to tree-structural models or models with low treewidths. I would expect the authors to discuss this limitation of their algorithm. I have several reservations regarding the experimental section for which the authors’ may wish to reply. First, only two prediction problems are evaluated; both with low treewhidth, a condition which suits the AGM more than its competitors and does not reflect most prediction problems. Second, different loss metrics are used for each of the problems. Third, while results for the first problem are reported as an average over 20 data splits, no details about the methodology of the second experiment are given. In addition, since no statistical significance was reported for the results of the latter experiment, I assume these results are reported on the entire data set. Fourth, instead of performing the parametric t-test to compare several algorithms on multiple (20) data sets, a non-parametric test (e.g., Friedman’s test with a post-hoc test) should have been exercised. Thus, whether the results in Table 1 are significant or not is due to a suitable test being exercised. Fifth, the AGM shows no advantage over its competitors for the two prediction tasks tested, and thus I would solicit the authors to discuss the method benefits, practicality, and generalizability to non-tree problems. Major issues Not all terms are defined (see e.g., Section 2.1 and 2.2) The authors claim: “This two-stage prediction approach can create inefficiencies when learning from limited amounts of data since optimization may focus on accurately estimating probabilities in portions of the input space that have no impact on the decision boundaries of the Bayesian optimal action” – but this claim is controversial wrt the author selection in a generative model (i.e., graphical models) for prediction. The authors are asked to motivate their selection in the graphical model that required applying complex mathematical transformations in order to make it a prediction model. Minor issues 56 – argmax vs argmin? Include directed graphical models (Bayesian networks) in you your survey of probabilistic graphical models to complete it. References: Capitalize “Bayesian”, conference/journal names, Arc; decide on a consistent use of author and journal names Quality: good Clarity: average Originality: average Significance: poor The authors response has raised my opinion about the experimental part of the paper and thus also about the paper itself.

Reviewer 3



The paper proposes a new supervised learning approach for graphical models that differs from structured output SVMs and from maximum likelihood learning. Given a loss function, the task is formulated as a minimax problem that searches the best inference strategy for the worst model among all probability distributions that have the same unary and pairwise marginals as the empirical distribution given by the training data. It is shown that the saddle point problem is tractable for acyclic graphical structures and graphs with low tree-width. Two simple(?) experiments demonstrate the superiority of the approach over structured output SVMs and maximum likelihood learning. Paper strengths: The proposed approach is to my knowledge novel and highly interesting. It can be seen as a variant of Bayesian learning over an infinite model class without explicit parametrisation and without the requirement to fix a prior distribution on it. At the same time it enjoys tractability for models on acyclic graphs and additive loss functions. This theoretical paper is well structured and by and large well written. Paper weaknesses: This highly theoretical paper his hard to read (at least for me). The vast number of auxiliary quantities and notations makes it hard to follow the derivation steps. I would have appreciated a simple example demonstrating the main ideas and steps of the approach. The description of the two experiments is somewhat too short - it requires the reader to consult quite a few papers to get a clear understanding of the tasks and the used models. Further comments/questions: When trying to gain more insight into the proposed method I have noticed that the inner constrained minimisation task in Definition 1 is in fact a linear task and can be replaced by its dual task. This seems to lead to the following (somewhat simplified) interpretation: The optimal inference minimises the energy of a graphical model with "loss augmented" unaries, whose parameters are chosen such that the unary and pairwise statistics taken over the optimal labellings for the training data inputs (x) is equal to the same statistics on the complete training data (x,y). I would appreciate if the authors could comment on this. Overall, this is a highly interesting theoretical paper that proposes a novel (supervised) learning approach for graphical models.